# Validation of a Sensor-Based Dynamic Ski Deflection Measurement in the Lab and Proof-of-Concept Field Investigation

**DOI:** 10.3390/s22155768

**Published:** 2022-08-02

**Authors:** Christoph Thorwartl, Josef Kröll, Andreas Tschepp, Helmut Holzer, Wolfgang Teufl, Thomas Stöggl

**Affiliations:** 1Department of Sport and Exercise Science, University of Salzburg, Schlossallee 49, 5400 Hallein/Rif, Austria; josef.kroell@sbg.ac.at (J.K.); wolfgang.teufl@sbg.ac.at (W.T.); thomas.stoeggl@sbg.ac.at (T.S.); 2Joanneum Research Forschungsgesellschaft mbH, Franz-Pichler-Straße 30, 8160 Weiz, Austria; andreas.tschepp@joanneum.at; 3Atomic Austria GmbH, Atomic Strasse 1, 5541 Altenmarkt, Austria; helmut.holzer@atomic.com; 4Red Bull Athlete Performance Center, Brunnbachweg 71, 5303 Thalgau, Austria

**Keywords:** bending sensors, flexion, PyzoFlex, ski bending, ski deflection

## Abstract

Introduction: Ski deflection is a performance-relevant factor in alpine skiing and the segmental and temporal curvature characteristics (m^−1^) along the ski have lately received particular attention. Recently, we introduced a PyzoFlex^®^ ski deflection measurement prototype that demonstrated high reliability and validity in a quasi-static setting. The aim of the present work is to test the performance of an enhanced version of the prototype in a dynamic setting both in a skiing-like bending simulation as well as in a field proof-of-concept measurement. Material and methods: A total of twelve sensor foils were implemented on the upper surface of the ski. The ski sensors were calibrated with an empirical curvature model and then deformed on a programmable bending robot with the following program: 20 times at three different deformation velocities (v_slow_, v_medium_, v_fast_) with (1) central bending, (2) front bending, (3) back bending, (4) edging left, and (5) edging right. For reliability assessment, pairs of bending cycles (cycle 1 vs. cycle 10 and cycle 10 vs. cycle 20) at v_slow_, v_medium_, and v_fast_ and between pairs of velocity (v_slow_ vs. v_medium_ and v_slow_ vs. v_fast_) were evaluated by calculating the change in the mean (CIM), coefficient of variation (CV) and intraclass correlation coefficient (ICC 3.1) with a 95% confidence interval. For validity assessment, the calculated segment-wise mean signals were compared with the values that were determined by 36 infrared markers that were attached to the ski using an optoelectrical measuring system (Qualisys). Results: High reliability was found for pairs of bending cycles (CIM −0.69–0.24%, max CV 0.28%, ICC 3.1 > 0.999) and pairs of velocities (max CIM = 3.03%, max CV = 3.05%, ICC 3.1 = 0.997). The criterion validity based on the Pearson correlation coefficient was r = 0.98. The accuracy (systematic bias) and precision (standard deviation), were −0.003 m^−1^ and 0.047 m^−1^, respectively. Conclusions: The proof-of-concept field measurement has shown that the prototype is stable, robust, and waterproof and provides characteristic curvature progressions with plausible values. Combined with the high laboratory-based reliability and validity of the PyzoFlex^®^ prototype, this is a potential candidate for smart ski equipment.

## 1. Introduction

The way a ski moves on the snow surface provides information about the performance and quality of a turn. A ski that slides sideways across the snow surface during forward motion is called skidding. In contrast, a carving turn is defined by minimal or no lateral displacement of the ski relative to the track, and, therefore, a point along the ski edge follows the path of the proceeding one [1,2,3,4]. To characterize a carving turn, the term “turn radius” has been established, which is defined by side-cut and ski deflection [5,6]. While carving, the turn radius can be changed by the manipulation of the ski deflection based on alterations in the applied skiing technique (e.g., applied forces, edge angle, body position). However, it is an oversimplification to assume a homogeneous deflection behavior (=constant radius) along the ski; rather, temporal and segmental differences in the curvature ($w^{\prime \prime }$) occur depending on the skiing technique and the quality of a turn [3,7]. Accordingly, it is not possible to deduce the deflection behavior based on the turn radius; instead, the $w^{\prime \prime }$ characteristic along the ski is decisive.

The most common method to measure ski deflection under standardized laboratory conditions is the three-point bending test [8,9,10,11,12,13,14]. For analyzing the ski deflection in the field, only strain gauges [6,15,16,17,18,19,20,21,22,23] have been used. The strain gauge-based approaches have not been tested for reliability or validity from a scientific point of view, and consequently the studies are limited for proof-of-concept investigations. In contrast, a novel PyzoFlex^®^ prototype has already been tested for reliability and validity in a quasi-static setting with a high-precision laser measurement system [7]. A curvature model was developed and the calibrated data provided both good accuracy (1.33 × 10^−3^ m^−1^) and high precision (4.14 × 10^−3^ m^−1^) [7]. However, no field measurements have been performed with this novel prototype so far.

The calibration of bending sensors is commonly performed in a quasi-static laboratory bending simulation. The resulting bending line of the deformed ski can be quantified by the vertical displacement [8,9,10,11,12], the angular deformation [13], or the curvature ($w^{\prime \prime }$) [7,14]. Those values can accordingly be used for sensor calibration. With this procedure, it is assumed that sensors that are calibrated in the quasi-static setting will measure correctly also during dynamic deflections in the field. There are three studies that have gone a step further and not only considered the quasi-static approach, but also verified the sensor signal in dynamics. In the study of Yoneyama et al. (2008), a free vibration measurement was performed, demonstrating good agreement between the signal from the strain gauges and the external inductive displacement transducer [18]. However, the method was limited to a qualitative description and the free vibration is a dynamic process, which is not very similar to the real ski-specific deflections. In the work of Thorwartl et al. (2022), the PyzoFlex^®^ prototype showed reliable and plausible curvature characteristics in a dynamic oscillating bending situation, although the measured curvature was only estimated and not compared with a gold standard measuring device [24]. To the best of our knowledge, the study of Adelsberger et al. (2014) is the only one which performed a validation in the field. The center of mass (CoM) turn radii that was obtained from a differential Global Navigation Satellite System was compared with data from strain gauges. The root mean square error between the radii that were calculated from deflection measurements and CoM radii was on average 1.26 m [6]. Moreover, it was pointed out that the radii were smaller relative to the track radii at the tip of the ski. This indicates an inhomogeneous deflection behavior of the ski, but only the total turn radius and not segment-specific radius differences were validated. Individual sensors were consequently not analyzed to validate the ski deflection.

None of the prototypes that were described above were subjected to a reliability or validity assessment in a dynamic ski-specific deformation setting. Validating the deflection in the field is complex, since we do not know of any gold standard measuring instrument for deflection detection on snow. Therefore, it is appropriate to perform the dynamic validation in the laboratory by deforming the ski using a programmable bending robot applying deformations that are similar to skiing (e.g., cyclic deformation, edging left and right). In addition to the influence of cyclic loads and combined deformations (torsion and bending), the ski would also have to be deformed with different excitation frequencies in order to determine and, if necessary, extend the scope of the deflection model.

By using such a laboratory setup, a gold standard instrument must be used that can detect ski deflection in three-dimensional space. The best accuracies of local position measurement technologies are provided by optoelectronic measurement systems (OMS) and, therefore, they are often considered as the “gold standard” in 3D motion capture [25,26]. The position accuracy of OMS is between 0.01 and 0.77 mm, depending on the capture system and experimental setup (e.g., the number of markers and cameras, marker size, size of the volume to be captured and distances between the cameras) [26,27,28,29,30,31,32]. However, the accuracy of the $w^{\prime \prime }$ detection can only be estimated to a limited extent from the local position accuracy. Based on measurements with a calibration profile it was shown that no valid $w^{\prime \prime }$ can be detected at a marker distance of <50 mm and only conditionally at 110 mm [33]. At a marker distance of 170 mm, valid curvatures can be detected since the maximum average difference was 1.11 × 10^−3^ m^−1^ in the static situation and −3.83 × 10^−3^ m^−1^, in the dynamic situation [33]. In only one study, the radius of curvature was estimated using an OMS for bending stiffness determination of alpine skis, but the $w^{\prime \prime }$ was not calculated exactly [12].

Since the recently published study of the novel PyzoFlex^®^ ski prototype was restricted to quasi-static setting, the purpose of the current study was to verify and extend the curvature model with respect to dynamic deflection. Therefore, the aims of this study were to (a) present an enhanced PyzoFlex^®^ ski prototype for curvature measurements on snow, (b) test the reliability and validity of the sensor-based $w^{\prime \prime }$ calculation by applying deformations that were similar to skiing in a dynamic setting, and (c) perform a proof-of-concept field measurement.

## 2. Materials and Methods

### 2.1. Further Developed PyzoFlex^®^ Prototype for the Detection of the Local Ski Bending Curvature

An improved prototype was developed based on the PyzoFlex^®^ ski demonstrator, which has already been presented for laboratory measurements [7,24]. PyzoFlex^®^ sensors (www.pyzoflex.com (accessed on 1 July 2022), Joanneum Research Forschungsgesellschaft m.b.H, Franz-Pichler Str. 30, 8160 Weiz, Austria) are generally made from the ferroelectric copolymer P(VDF-TrFE) (poly(vinylidene-trifluoroethylene)) and are, therefore, inherently piezo- and pyro-electric [33]. For the sensor system that was used in this work, 24 sensor elements, divided into three foil segments (2 × 9 and 1 × 6 elements), were fabricated using screen printing on a single substrate. The segments were then cut out with a Trotec laser cutter and laminated to the surface of the ski (Atomic Redster G7; length: 1.82 m; radius: 19.6 m) with an adhesive tape from 3M™ series (VHB™ 5907F). Since the sensor foil is applied to the adhesive foil with the printed side facing downwards, this adhesive layer also acts as a seal against water penetration. Only the sensor flag at the edge of the foil is not adhered to the tape to enable the electrical connection to the readout electronics (Figure 1a).

For establishing contact between the sensor foils and the measurement data acquisition device (DAQ), the printed silver conductor lines were used with a foil connector from Amphenol (FCI Clincher™ Flex Connectors). To avoid short circuits at the silver contacts due to water penetration, the three contact points per ski were separately housed and sealed with silicone. All three sensor cables were merged at one point into the DAQ. In comparison to our former system [7], explicit emphasis was placed on the robustness of the system. Previously used ribbon cables were replaced by more robust sheathed and shielded multicore cables. One DAQ unit was installed per ski, which is capable of processing 16 of the 24 sensor channels in parallel. In addition, each evaluation unit has a so-called synchronization unit. On the one hand, this makes it possible to start and stop measurements, and on the other hand it enables synchronizing the data acquisition with the video recording by switching a LED that can be filmed by an external camera. The trigger time (i.e., the switching time of the LED) is stored in the database together with the sensor data and allows the synchronization with the video stream afterwards. Basically, the system can be operated in two modes. Either the data stream is transmitted instantly via Wi-Fi, displayed on a PC, and stored for further processing, or the data is stored on an on-board SD card and is read out and processed afterwards. Both the laboratory and field measurements were performed in the mode in which the data were recorded on the SD card. Figure 1b shows the complete PyzoFlex^®^ demonstrator at the first system check on snow.

In order to derive the local ski bending curvature from the measured charge response Q(*t*) of the sensor foils, our previously published calibration method was applied [7]. A charge amplifier circuit with a feedback capacity of $C_{f}=20\;\mathrm{nF}$ converts Q(*t*) into a corresponding output voltage $u_{a}\left(t\right)$. The mean curvature, $\bar{w^{\prime \prime }}$, is proportional to $u_{a}\left(t\right)$ and given by Equation (1):(1)$$\bar{w^{\prime \prime }}\left(u_{a}\right)=\frac{-C_{f}}{A\cdot v_{13}\cdot a\cdot P_{r}\cdot \zeta }\cdot \left(u_{a}-u_{0}\right)=\left(k\cdot u_{a}+d\right)\cdot f_{d}$$
where A is the interface area between the electrode and the ferroelectric material, $v_{13}$ is the effective Poisson’s ratio, a is a material constant close to unity, $P_{r}$ is the remnant polarization resulting directly from the poling process, *ζ* denotes the distance to the neutral axis, and $u_{0}$ is the voltage baseline offset. Since *ζ*(x) is not available because it depends on the internal structure and the elastic properties of the rather complex ski construction, a calibration procedure is required for each combination of ski design, sensor design and alignment. Therefore, a two-point calibration was performed using a bending machine with a high-precision laser measurement system [7]. A linear regression model was established for each individual sensor and the calibration factors (slope *k* and the intercept *d*) were used to derive the segment’s mean curvature $\bar{w^{\prime \prime }}$ from the voltage $u_{a}$. Furthermore, based on preliminary investigations, a dynamic correction factor ($f_{d}\;$= 0.78), which is constant for all the sensors, was included in Equation (1). This is necessary to compensate the relaxation effect that occurs in the quasi-static calibration process of the binding machine due to the viscoelastic behavior of the adhesive tape (Figure A3 in Appendix A). Moreover, during calibration the ski is not deformed from a neutral, but from a slightly pre-stressed initial position. The dynamic calibration factor also takes corrective action here.

### 2.2. Reliability and Validity Assessment in the Laboratory

#### 2.2.1. Experimental Setup

An industrial articulated-arm robot (IRB 6400R, ABB AG, Brown-Boveri-Strasse 5, 8050 Zurich, Switzerland) was used to deform the PyzoFlex^®^ technology-based ski prototype in a standardized dynamic setting. The robot’s working space is thermally sealed off from the environment, which means that the temperature (T = 19 °C) in this area can be continuously adjusted. The robot’s base frame, carousel, and weight-balancing system are located outside the thermally adjustable working space.

The ski was fixed to the robot arm by a test shoe. Underneath the ski was a frame that was made of aluminum profiles with rotatable synthetic rollers as supports at the front and the back of the ski. In the neutral position (NP), the ski rests unloaded on the synthetic rollers. The robot was programmed to cyclically deform the ski through five deformation modes. An overview of the deformation sequence that was described by the movement of the robot arm coordinate system (x_R_, y_R_, z_R_) is shown in Figure 2. In the first mode, a central bending (CB) in the z_R_-direction was performed, followed by a bending of the front (FB) and rear ski (BB) in y_R_-z_R_ direction. Deformation modes 2 and 3 are intended to simulate forward and backward leaning of the skier. In modes 4 and 5, the ski is first edged to the left (EL) and then to the right (ER) by a lateral movement in the x_R_-z_R_-direction while tilting around the y_R_ axis. The edging and the resulting combined load of torsion and bending is supposed to simulate a carving turn. After ER, the cycle starts again from the beginning with CB. The ski returns to the NP between the modes, except for the transition from CB to FB. For the study, the entire deformation program (CB, FB, BB, EL, ER) was run 22 times at three different cycle times (CT_1_ = 16 s, CT_2_ = 12 s, CT_3_ = 8.5 s), changing only the deformation velocity (v_slow_, v_medium_, v_fast_) of the robot arm but not the deflection amplitudes. A PyzoFlex^®^ prototype has already been tested for reliability on a pneumatic test bench over 90 cyclic oscillating repetitions [24]. For the reliability assessment, the first ten peak values of the sensor signal were compared with the middle and last ten peak values. Since a very reliable behavior is shown over 90 repetitions, the amount of 22 repetitions seems to be sufficiently large.

Reliability and validity assessments were performed with the instrumented ski that was described in Section 2.1, excluding twelve sensors (the front six sensors and the middle row). The corresponding sensors were not considered because, on the one hand, the number of sensor inputs of the DAQ was limited to 16 channels and, on the other hand, no comparison values from the gold standard measuring device were available for the validity assessment in the medial and anterior ski segments. To detect the three-dimensional ski deflection, an OMS (=gold standard) from Qualisys (Oqus 7+, Qualisys AB, Kvarnbergsgatan 2, 411 05 Göteborg, Sweden) with an acquisition frequency of 240 Hz was used. A total of eight active infrared cameras detected the markers within a capture volume of 3 m × 2 m × 1.5 m. The positions of the cameras and the description of the components can be found in a previously published Technical Note [33]. There were three plastic marker bases that were attached to each of the 12 sensors (L_1_, …, L_6_ and R_1_, …, R_6_) using an adhesive tape of the 3M™ VHB™ series (F9460PC) with a thickness of 0.05 mm. For this purpose, a 3D-printed template (Figure A1) was fabricated to ensure that the bases were applied as standardized as possible. In total, 36 infrared markers with a diameter of 6.5 mm and an internal M4 thread were screwed onto the plastic bases (Figure 3).

#### 2.2.2. Data Processing–Motion Capture System

Based on the fact that no valid curvatures can be detected with Qualisys over a sensor with a length of 100 mm [33], the ski is divided into four bending segments (left rear segment (LBS), right rear segment (RBS), left front segment (LFS), and right front segment (RFS)) with a length of 340 mm (between two sensors is a gap of 20 mm). According to Figure 3, nine infrared markers are located in one bending segment. In principle, only three points are required to uniquely define $w^{\prime \prime }$ of a bending segment in three-dimensional space. However, to improve the signal-to-noise ratio and to minimize fluctuations in the curvature signal, a circle of the form:(2)$$C:\;\vec{x}=\vec{M}+R\cdot \cos\left(t\right)\cdot \vec{u}+R\cdot sin\left(t\right)\cdot \vec{v}\;$$
was fitted through nine measurement points using the least squares method [33]. $\vec{M}$ corresponds to the location vector of the center, R is the radius of the fitted circle, t refers to the free parameter of the parameter form and is between 0 and 2π, and the orthogonal unit vectors $\vec{u}$ and $\vec{v}$ lie in the plane of the circle. A customized algorithm in MATLAB (R2018B, MathWorks, Natick, MA, USA) was applied to perform the fitting routine based on Equation (2) and derive R. The mean curvature, $\bar{w^{\prime \prime }}$, can be calculated from the reciprocal of *R* using Equation (3):(3)$$\bar{w^{\prime \prime }}=\frac{1}{R}\;$$

Since the infrared markers were instrumented on the top side of the ski, but the calibration of the PyzoFlex^®^ sensors was based on a high-precision laser measurement system (LK-H157, Keyence AG, Japan) on the bottom side, a discrepancy would arise when comparing the detected $w^{\prime \prime }$ from the sensor with the captured $w^{\prime \prime }$ from Qualisys. On the one hand, the $w^{\prime \prime }$ is geometrically larger on the upper side than on the lower side, and on the other hand, even slight differences in the instrumentation of the infrared markers can lead to a $w^{\prime \prime }$ offset in the NP. Therefore, the Qualisys system was calibrated preliminarily on a bending machine with a high-precision laser measurement system (Figure A2a) using Equation (4):(4)$$\bar{{w_{bottom}}^{\prime \prime }}=k_{laser}\cdot \bar{w^{\prime \prime }}+d_{laser}\;$$

There was a high linear correlation (r > 0.99) between the measured $w^{\prime \prime }$ from the laser and the detected $w^{\prime \prime }$ from the Qualisys system (Figure A2b). Equation (4) was, therefore, applied to calculate the segmental mean curvatures ($\bar{{w_{LBS}}^{\prime \prime }}$*,*$\;\bar{{w_{RBS}}^{\prime \prime }}$*,*$\;\bar{{w_{LFS}}^{\prime \prime }}$*,*
$\bar{{w_{RFS}}^{\prime \prime }}$) at the bottom of the ski for the associated segments (LBS, RBS, LFS, RFS).

#### 2.2.3. Data Processing—PyzoFlex^®^ Sensor System

All PyzoFlex^®^ data that were recorded during the measurement were filtered and linear drift-corrected using MATLAB. In particular, a 2nd order Butterworth high-pass filter with a cut-off frequency of 0.2 Hz was implemented to remove temperature-related signal fluctuations due to the pyroelectric effect from the raw data. Afterwards the sensor signals were calibrated as described in Section 2.1. The first and last cycles were not considered for the analysis based on the ringing artifacts by the filter in the border area, therefore 20 cycles remain for the analysis. For the validity assessment, segment-wise mean signals (LBS: (L_1_ + L_2_ + L_3_)/3, RBS: (R_1_ + R_2_ + R_3_)/3, LFS: (L_4_ + L_5_ + L_6_)/3 and RFS: (R_4_ + R_5_ + R_6_)/3) are formed, since a valid curvature can only be detected over a length of three sensors [33]. Since the maximum values of the sensor signals are decisive for the data processing, these values were determined by a custom peak detection algorithm in MATLAB.

#### 2.2.4. Statistical Analysis

To determine the dynamic performance of PyzoFlex^®^ technology-based sensors, the aspects of instrument reliability (within the PyzoFlex^®^ sensor system) and criterion validity (between systems) were examined. For the statistical analysis, 12 sensors (L_1_ to L_6_ and R_1_ to R_6_) with the corresponding peak values (in m^−1^) at CB, FB, BB, EL, and ER were considered.

To assess the reliability, data were examined statistically (Shapiro–Wilk test) for normal distribution. Changes in the mean (CIM) for three dynamic situations expressed by the deformation velocity (v_slow_, v_medium_, v_fast_) were analyzed using a paired samples *t*-test (level of significance *p* < 0.05). For the assessment of relative test-retest reliability, the typical error of measurement represented as the coefficient of variation (CV) and the intraclass correlation coefficient (ICC 3.1) were utilized. ICC 3.1 was interpreted as unacceptable (≤0.74), good (0.75–0.89), and excellent (≥0.9) [34]. Successive pairs of trials (cycle 1 with cycle 10, cycle 10 with cycle 20) were considered for CIM, CV, and ICC 3.1. In addition, an analogous test-retest reliability between different deformation velocities (v_slow_ vs. v_medium_ and v_slow_ vs. v_fast_) was analyzed by considering the peak values of cycle 1, 10, and 20.

Pearson’s correlation coefficient was used for criterion validity between PyzoFlex^®^ and the Qualisys system. The magnitude of correlations was graded as follows: r < 0.45, impractical; 0.45–0.70, very poor; 0.70–0.85, poor; 0.85–0.95, good; 0.95–0.995, very good; and >0.995, excellent [35]. To describe the agreement between the sensor system and the Qualisys measurement method, a Bland–Altman plot was employed [36]. The accuracy, expressed by the systematic bias, and the precision, expressed by the standard derivation (SD), were evaluated to assess the limits of agreement (LoA) (±0.96 × SD). All the metrics were calculated using MATLAB and a custom spreadsheet [35], except for the paired samples *t*-test statistics, for this IBM SPSS Statistics V.26.0 (SPSS Inc., Chicago, IL, USA) was used.

### 2.3. Proof-of-Concept Field Measurement

The proof-of-concept field measurement was carried out on 2 March 2021 in the Petersbründel ski area (Austria). The experimental setup is illustrated in Figure 4. The test person (A-level ski instructor) performed 24 carving turns (12 left turns and 12 right turns) with long radii within a defined measurement corridor. A follow cam and a body cam were used to determine the turn switch points (TSPs) of the carving turns. At the beginning and end of the run, the self-built synchronization unit was used to perform the sync event to synchronize the PyzoFlex^®^ data with the video data.

Only the left row of the right ski was used for the descriptive analysis. The recorded sensor signals were filtered and linearly drift-corrected analogously to the laboratory setting and additionally processed with a custom zero-update algorithm using MATLAB. Further, the data were time normalized over a left-right turn combination and then the mean $w^{\prime \prime }$ ± standard error (SE) was calculated.

## 3. Results

### 3.1. Descriptive Report of Lab Data

Figure 5 shows the detected $w^{\prime \prime }$ as a result of the sequence of five deformation modes (CB, FB, BB, EL, and ER) at v_fast_. For the illustration of the 12 sensors, the mean value ± SD (in m^−1^) over 20 repetitions was calculated for each measuring point. A differentiation is made between the left (L_1_ to L_6_) and right sensor rows (R_1_ to R_6_), showing all the rear sensors in blue (L_1_, L_2_, L_3_; R_1_, R_2_, R_3_) and the front sensors in green (L_4_, L_5_, L_6_; R_4_, R_5_, R_6_). The five deformation modes result in four peaks per sensor, since only the front or rear ski were deformed in the case of FB or BB. The sensors in the rear segments 2 and 3 are activated the most, followed by the sensors in segments 6, 5, and 4. The sensors in the first segment are partially above the support and are the least curved.

### 3.2. Instrument Reliability and Accuracy

Table 1 shows test-retest reliability results for both repetition pairs (cycle 1 vs. cycle 10 and cycle 10 vs. cycle 20) at three different deformation velocities (v_slow_, v_medium_, and v_fast_). The CIM over all the comparisons ranged between −0.69% and 0.24%. The maximum CV value of 0.28% can be noted when comparing cycle 1 with 10 at v_medium_. ICC 3.1 was very close to 1 (ICC 3.1 > 0.999, *p* < 0.001) for all cases and significant differences are detectable in four of the six comparisons.

Table 2 shows the test-retest reliability results between different velocities (v_slow_ vs. v_medium_ and v_slow_ vs. v_fast_). The CIM is greater at a higher velocity difference (3.03%) than when comparing v_slow_ with v_medium_ (1.85%). The maximum CV value is 3.05% and the ICC 3.1 is greater than 0.997. A significant difference can be found in both comparisons.

### 3.3. Criterion Validity

Figure 6 shows the signal of the PyzoFlex^®^ ski in red and the gold standard system (Qualisys) in blue over three bending cycles at v_medium_. In the BB situation, the biggest difference between Qualisys and PyzoFlex is obviously seen in the peaks of LBS and RBS. The Qualisys data in the grey marked area cannot be judged, as valid results from Qualisys can only be expected from $w^{\prime \prime }$ of 0.13 m^−1^ upwards [33].

In Figure 7a, data from the PyzoFlex^®^ sensor system were correlated to the criterion instrument (Qualisys system) to determine validity. The slope of the function is 1.03 and the coefficient of determination R² is 0.97. Figure 7b shows the Bland–Altman plot with the systematic bias of 0.003 m^−1^ and the LoA (−0.096 m^−1^ and 0.090 m^−1^).

### 3.4. Proof-of-Concept Field Measurement

Figure 8 shows the time-normalized $w^{\prime \prime }$ (mean ± SE) of 24 carving turns with long radii. The TSP from a right to a left turn is at 0% and 100% and from a left to right turn at 50%. The rear sensors are shown in blue and the front sensors in green. The sensor behind the binding plate (L_3_) shows the highest peak with a value of about 0.8 m^−1^.

## 4. Discussion

The objective of this study was to present an improved PyzoFlex^®^ ski prototype for the detection of $w^{\prime \prime }$, to evaluate its test-retest reliability and criterion validity in a dynamic laboratory setting, and to perform a proof-of-concept field measurement. The PyzoFlex^®^ prototype can be considered as an accurate and precise instrument, as the data showed a very good agreement with the reference system (Qualisys) and produced plausible $w^{\prime \prime }$ characteristic on the snow.

The results have shown that the ski prototype provides highly reliable data. Within the velocities, the max CIM was 0.69%, the max CV was 0.28%, and the ICC 3.1 was found to be consistently close to 1 (all ICC 3.1 > 0.999), indicating excellent reliability (Table 1) [34]. Comparing the values between the velocities, both the CIM and CV are slightly higher in the range of 1.84% to 3.05%, with the ICC 3.1 again very close to 1 (Table 2). Out of the total eight comparisons, there were six significant differences, with the largest difference evident for v_slow_ vs. v_fast_ (CIM = 3.03%, *p* < 0.001). When a comparable reliability test was carried out with the Qualisys data for v_slow_ vs. v_fast_, a small but significant difference (CIM = −0.14%, *p* < 0.001) is also noticeable. However, the small but significant difference argues for the accuracy of the PyzoFlex^®^ system, as the differences can be at least partially explained by the Qualisys data [7].

In terms of validity, there was also good agreement between the calculated $w^{\prime \prime }$ of the sensor signal and the captured $w^{\prime \prime }$ by the Qualisys system. The magnitude of the linear relationship between the two systems, expressed by the correlation coefficient of r = 0.985, is very good (Figure 7a) [35]. Moreover, a high accuracy was shown since the systematic bias was only −0.003 m^−1^. The precision of the signal denoted by the SD (0.047 m^−1^) and LoA (−0.096 m^−1^ and 0.090 m^−1^) is consistent with the other results and indicates high validity (Figure 7b). However, it is evident that the accuracy for the LBS and RBS in the BB situation is comparatively low (Figure 6). This could be since the posterior segments have the greatest $w^{\prime \prime }$ and thus the relaxation effect of the tape is more pronounced, making the PyzoFlex^®^ $w^{\prime \prime }$ appear to be smaller. As can be seen in Figure 6, there is hardly any difference between the parallel mounted sensors. In particular, the differences in EL and ER were expected, as this represents a combined load of torsion and bending, and this should be evident in the differential signal of the parallel mounted sensors. However, this result is consistent with the finding that ski deflection consists mainly of bending and that torsional deformation has only a minor influence on the shape of the running surface [18].

In previous studies, little attention was paid to ski $w^{\prime \prime }$ except for our two studies with the PyzoFlex^®^ prototype [7,24]. First, the reliability of the sensors in the presented experimental setup is comparable to those in the quasi-static setting, as the CIM is −1.41%, the max CV is 1.45%, and the ICC 3.1 is >0.961 [7]. Second, the reliability assessments are also comparable to the results of a oscillating bending measurement (CIM = 0.20%, CV = 0.63% and the ICC 3.1 > 0.997) [24]. However, most studies have focused on sensor-based detection of the deflection line w(x), but not on the analysis of $w^{\prime \prime }\left(x\right)$, and further, no reliability and validity tests have been performed, so that no comparison with previously used systems and prototypes is possible.

The proof-of-concept measurement has shown on the one hand that the prototype is stable, robust, and waterproof and on the other hand that plausible $w^{\prime \prime }$ values can be measured. The $w^{\prime \prime }$ shows a characteristic pattern with the highest $w^{\prime \prime }$ between the TSPs (Figure 8). However, the rapidly fluctuating $w^{\prime \prime }$ within a turn is consistent with the findings that the ski shape changes rapidly within a short time period [18]. When comparing the front and rear segments, temporal differences can be observed, as the front sensors are activated at an earlier stage relative to the TSPs. Based on the analysis of the video data, this seems plausible since during TSPs often the rear end of the ski is in the air and the front end is pre-stressed (see middle image in Figure 8). According to the definition of a carving turn, that the rear section of the ski glides in the front groove that is formed by the front section [5], an earlier temporal activation of the front sensors seems to be reasonable. The rear segment is more deformed, which is consistent with both the measurements with the bending robot that was described above and the results of the prototypes that are based on strain gauges [16,17]. Furthermore, it is also reported that the effective bending radius of the ski in carving turns is about 10 m, while the true turn radius is 20 m [18]. This can also be confirmed in this work, as the max value of the mean $w^{\prime \prime }$ across all sensors (L_1_ + ,…, + L_6_)/6) is 0.19 m^−1^. This is equivalent to a radius of about 5.3 m, which is at least twice as large as the true turn radii.

The study that is presented has some limitations. First, single sensors with a length of 100 mm cannot be checked for validity, as this is outside the accuracy range of the 3D motion capture system (Qualisys) [33]. Averaging over three sensors is, therefore, a necessary procedure for the validity assessment. Since there is a distance of 2 cm between the sensors, two small sub-segments are not taken into account for $w^{\prime \prime }$ detection. Secondly, it was observed that the sensors that were calibrated in a quasi-static environment would not give valid results in a dynamic setting, so a correction factor ($f_{d}$) was included in the curvature model. This is primarily justified by the relaxation phenomena due to the viscoelastic behavior of the tape (Figure A3). Additional measurements have shown that, on the one hand, the relaxation effect is lower with thinner adhesive tapes (200 µm instead of the 1.1 mm version of the tape) and, on the other hand, larger sensor deflections can be observed due to the more direct connection. For the future, dynamic calibration appears to be a reasonable option. Therefore, it seems appropriate to develop a test bench with high-precision laser measurement systems on which the ski can be dynamically deformed and the single sensors calibrated. This would also have the advantage that, in contrast to Qualisys, $w^{\prime \prime }$ below 0.13 m^−1^ can also be recorded. Furthermore, it should be noted that due to the pyroelectric nature of the sensors, the temperature influence has not yet been sufficiently clarified. Basically, it is assumed that the low-frequency temperature changes can be filtered out with a high-pass filter. Measurements in a cold chamber are necessary to clearly determine the temperature-related influence.

The novel application needs to be progressively developed in different areas. In the current prototype, three sensor elements per ski communicate with the DAQ via two interfaces. In perspective, one large sensor element is to be used over the entire length of the ski so that the interfaces are reduced to one connection behind the binding plate. This has the advantage that the conductor paths run under the binding plate and the sensors can also be integrated in this segment. For easier handling, the Raspberry Pi-based DAQ should be replaced by a microcontroller-based system and screwed directly to the ski boot. Perspectively, an integration of the sensor foils into the ski construction would be advised to protect the sensors from mechanical damage and environmental influences.

Further in-depth snow measurements are required to investigate the equipment- and technique-specific ski-snow interactions. The knowledge that is gained can flow into various development and application fields. First, the ski deflection and behavior that was identified on the basis of the sensor activity, might serve as a potential application for injury analysis and prevention. In terms of product fitting, the PyzoFlex^®^ technology-based ski could help to find the appropriate ski, not only in terms of safety but also in terms of performance. For example, if the deflection of the ski is too low according to the measured $w^{\prime \prime }$, it seems appropriate to choose a ski with a lower bending stiffness. On the other hand, personalized feedback can be provided to improve the skiing technique and performance. Of particular interest is also the extent to which the segmental and temporal curvature characteristics differ depending on the performance level (high-performance athletes vs. amateur athletes) and the technique that is used (carving, parallel ski steering, snowplough steering, etc.). In order to shed light on the construct “ski deflection”, it seems reasonable to use additional measurement systems for snow investigations. Thus, plausibility checks, pattern recognition, and technical validation can be carried out. In terms of discriminant validity, correlations between different measures (such as edge angle or radial force) can provide revealing information. Furthermore, the indoor test methodology for ski development is to be designed more realistically based on the field findings in order to increase the safety and stability of product engineering.

## 5. Conclusions

Ski deflection is the result of a multifactorial interaction between the ski and the snow and has a crucial influence on skiing performance and on the quality of a turn. Up to now, there are no commercial applications that measure ski deflection during alpine skiing. The ski prototype that was presented here, based on PyzoFlex^®^ technology, opens a potential field of application for smart ski equipment, since segmental $w^{\prime \prime }$ can be reliably and validly detected. In addition to the possibility of a real-time feedback system, application in the field of product fitting, and injury analysis, conclusions can also be drawn about indoor testing methodology for ski development. For the future, however, further improvements of the prototype are planned (e.g., the integration of the sensor foils into the ski, enhancement and downsizing of the DAQ and implementation on the ski boot, reduction of the number of interfaces on the ski).

## Figures and Tables

**Figure 1 sensors-22-05768-f001:**
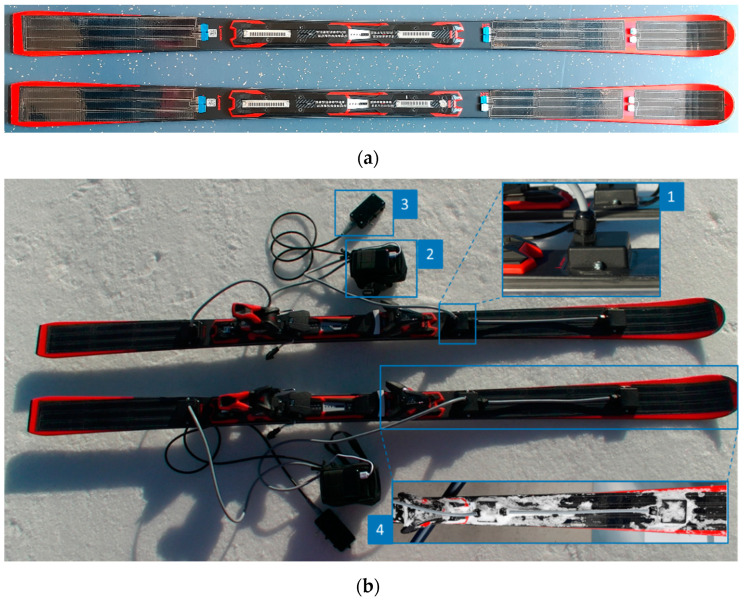
Ski prototype (**a**) after laminating the sensor foils and (**b**) after complete assimilation of all components. 1: Encased contact point that was sealed with silicone; 2: measurement data acquisition device (DAQ); 3: synchronization unit; 4: waterproof ski after system check.

**Figure 2 sensors-22-05768-f002:**
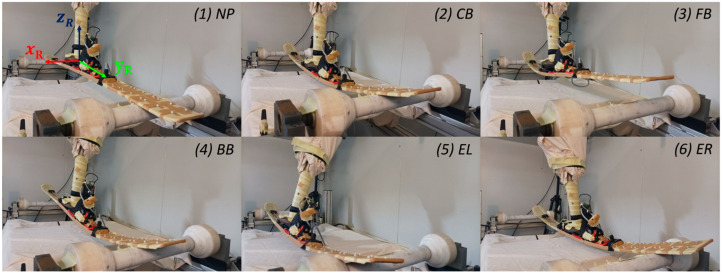
Experimental setup consisting of bending robot with the reference coordinate system (x_R_, y_R_, z_R_) and the PyzoFlex^®^ ski prototype that was equipped with the instrumented 3D markers. Deformation sequence of the ski on the bending robot: NP: neutral position; CB: central bending; FB: front bending; BB: back bending; EL: edging left; ER: edging right.

**Figure 3 sensors-22-05768-f003:**
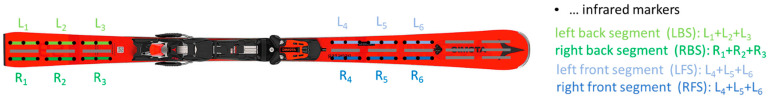
The PyzoFlex^®^ ski prototype with the corresponding left (L_1_, …, L_6_) and right (R_1_, …, R_6_) sensor row. There were three infrared markers (black dots) that were instrumented per sensor, resulting in a total of 36 markers. The segments were grouped into posterior (LBS: left back segment, RBS: right back segment) and anterior deflection segments (LFS: left front segment, RFS: right front segment).

**Figure 4 sensors-22-05768-f004:**
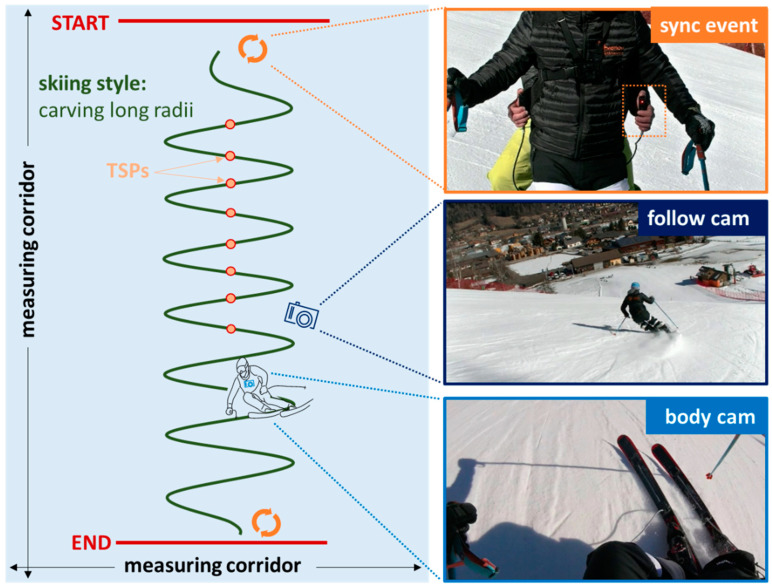
Experimental field setup. TSPs: Turn switch points.

**Figure 5 sensors-22-05768-f005:**
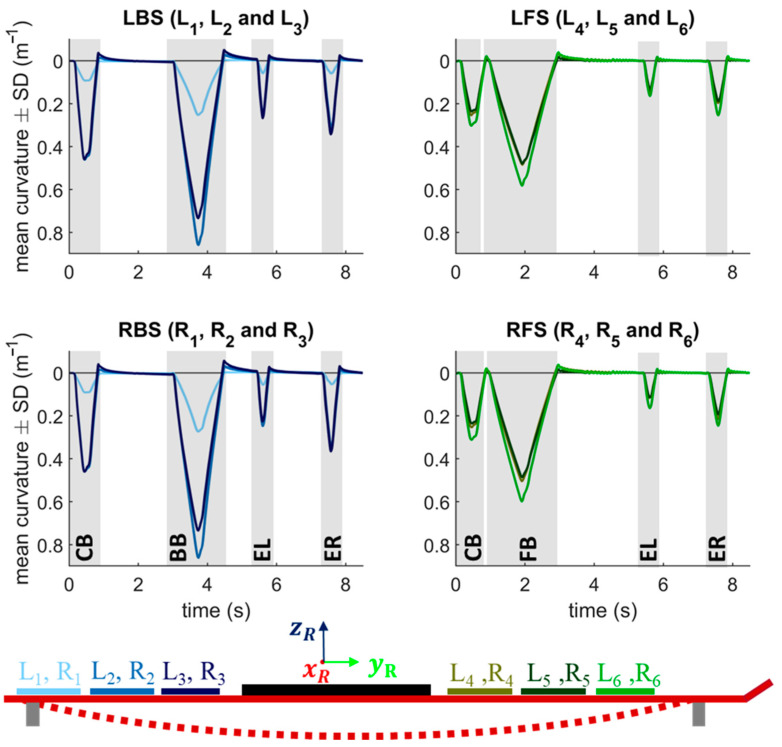
The PyzoFlex^®^ sensor signals (mean +/− standard deviation (SD)) over 20 cycles during dynamic loading with v_fast_. The rear sensors (R_1_, R_2_, R_3_, L_1_, L_2_, and L_3_) are shown in blue and the front sensors (R_4_, R_5_, R_6_, L_4_, L_5_, and L_6_) in green. The ski was deformed with a bending robot through five deformation modes (CB: central bend; FB: front bend; BB: rear bend; EL: edge left; ER: edge right).

**Figure 6 sensors-22-05768-f006:**
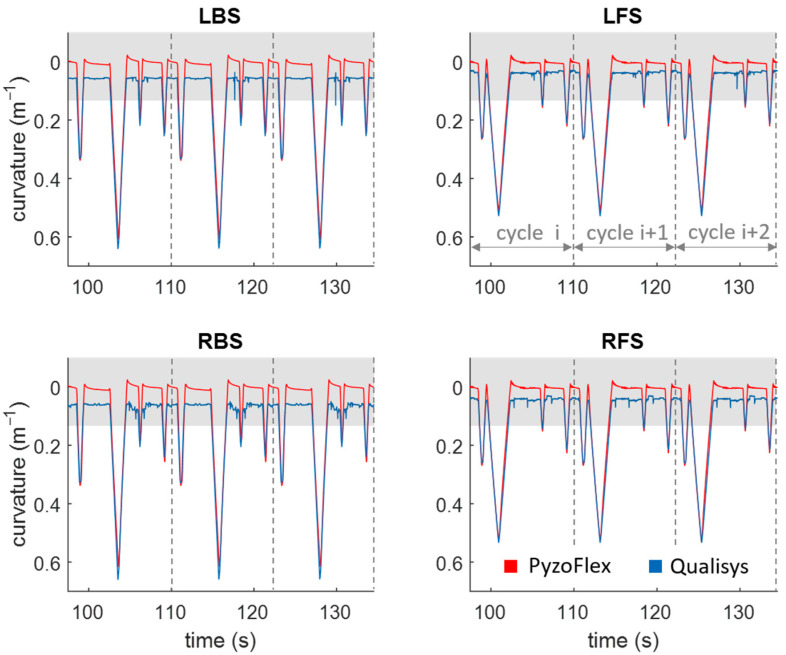
PyzoFlex^®^ signal in red and Qualisys (QTM) signal in blue over three cycles during dynamic loading at v_medium_. A differentiation is made between the posterior (LBS: left posterior segment, RBS: right posterior segment) and the anterior deflection segments (LFS: left anterior segment, RFS: right anterior segment).

**Figure 7 sensors-22-05768-f007:**
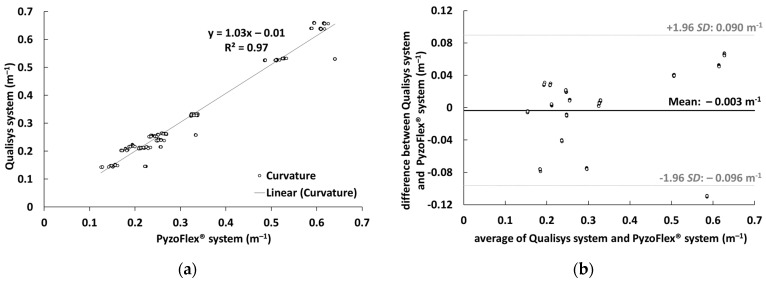
(**a**): Correlation between the curvature (m^−1^) that was measured by Qualisys system (criterion instrument) and PyzoFlex^®^ sensor system. (**b**): Bland–Altman plot showing the difference against the average of Qualisys system and PyzoFlex^®^ sensor system with limits of agreement (LoA) (dotted lines). SD: Standard deviation.

**Figure 8 sensors-22-05768-f008:**
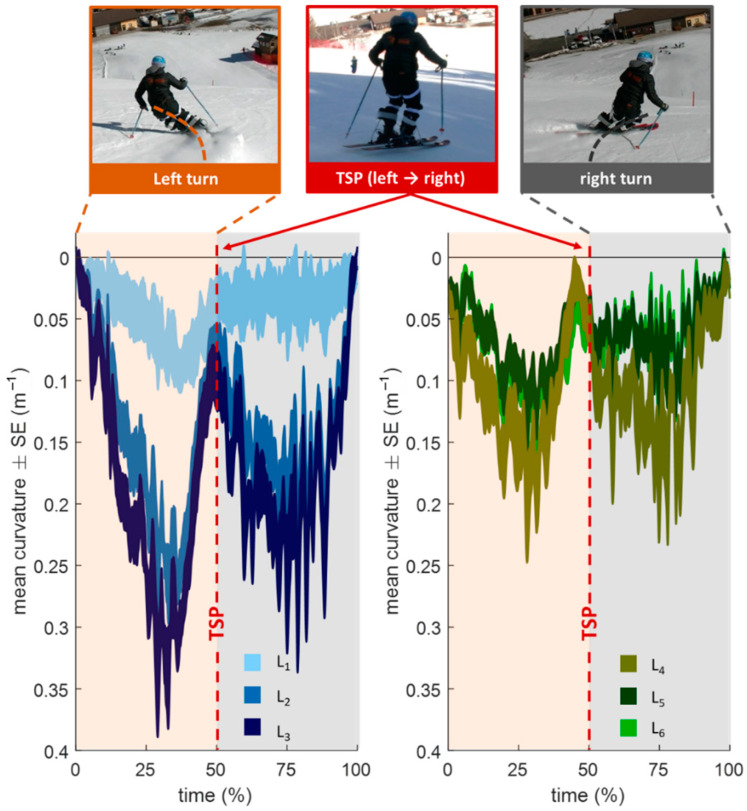
Result of the proof-of-concept field measurement with carving long radii (12 left and 12 right turns). The plot differentiates between the left and right turns. The rear sensors (L_1_, L_2_, L_3_) are shown in blue and the front sensors (L_4_, L_5_ and L_6_) in green tones. SE: standard error.

**Table 1 sensors-22-05768-t001:** Analysis of instrument reliability from the sensor system.

		*p* (*t*-Test)	CIM (%, 95% CI)	CV (%, 95% CI)	ICC 3.1 (95% CI)
v_slow_	cycle 1 vs. 10	0.010	0.09 (0.00–0.18)	0.26 (0.22–0.31)	0.999 (0.999–0.999)
	cycle 10 vs. 20	n.s.	0.01 (−0.06–0.08)	0.21 (0.18–0.26)	0.999 (0.999–0.999)
v_medium_	cycle 1 vs. 10	0.016	0.14 (0.05–0.24)	0.28 (0.24–0.34)	0.999 (0.999–0.999)
	cycle 10 vs. 20	n.s.	−0.01 (-0.08–0.07)	0.22 (0.19–0.26)	0.999 (0.999–0.999)
v_fast_	cycle 1 vs. 10	<0.001	0.24 (0.17–0.31)	0.21 (0.18–0.25)	0.999 (0.999–0.999)
	cycle 10 vs. 20	<0.001	−0.69 (-0.75–0.63)	0.17 (0.15–0.21)	0.999 (0.999–0.999)

CIM: change in mean; CI: confidence interval; CV: coefficient of variance; ICC 3.1: intraclass correlation coefficient.

**Table 2 sensors-22-05768-t002:** Analysis of the reliability of the instruments of the sensor system in respect of different deformation velocities.

		*p* (*t*-Test)	CIM (%, 95% CI)	CV (%, 95% CI)	ICC 3.1 (95% CI)
Cycle 1, 10 and 20	v_slow_ vs. v_medium_	<0.001	1.84 (1.25–2.43)	3.05 (2.74–3.38)	0.997 (0.996–0.998)
	v_slow_ vs. v_fast_	<0.001	3.03 (2.61–3.46)	2.18 (1.99–2.41)	0.999 (0.998–0.999)

CIM: change in mean; CI: confidence interval; CV: coefficient of variance; ICC 3.1: intraclass correlation coefficient.

## Data Availability

The data presented in this study are available on request from the corresponding author.

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
