# Peer review of "Validation of a Sensor-Based Dynamic Ski Deflection Measurement in the Lab and Proof-of-Concept Field Investigation"

_sensors, 2022, doi:10.3390/s22155768_

Round 1

Reviewer 1 Report

Some of the comments the authors can address to improve the quality of the paper:

1. Please notice that line 17  ...demonstrated high reliability and validity... is contradictory to line 428 ...cannot be checked for validity, ....

2. Line 28, please check the number of parentheses.

3. Line 326, I was wondering whether ICC 3.1 in the paper is Shrout-Fleiss reliability: fixed set (ICC(3,1)).

4. Line 328, why not Convert “CIM= -0.033” to a percent like other CIM results in the paper?

5. Line 403, check the sentence ...the ICC is 3.1 > 0.961”.

6. In the paper, application of the product is necessary, like injury analysis or comparison between high-performance athletes and ordinary athletes.

7. References are not novel enough, I recommend adding some recent papers.

Reviewer 2 Report

Comments to the manuscript with ID sensors-1826954 entitled: Validation of a sensor-based dynamic ski deflection measurement in the lab and proof-of-concept field investigation. This manuscript is about the validity of a ski sensors using a deflection. The manuscript must be rewrite due to the complexity in some paragraphs to understand clearly. Major changes are required.

Abstract:

Please, add in abstract the sections, Introduction, material and methods, results and conclusion. And specify clearly the aim of the research.

Line 19, please delete the (l1..L2…) this is confuse.

Introduction.

Lines 42-43: Is not clear the meaning of the sentences. Please rewrite. Is difficult understand. This happens in the introduction in some paragraphs. (eJ line 63…)

Material and methods

Can explain authors how calculate the amount of measurements to validate the sensors

Results: Ok

Discussion:

The discussion must be rewrite based in previous research. Some paragraphs looks like results.

The research has a lot of limitations. How can authors ensure the reliability of the research?

Conclusion:

Please rewrite the conclusion in 5 sentences.

Round 2

Reviewer 1 Report

All questions and recommendations have been addressed in the revised version. I recommend to accept this manuscript.

Reviewer 2 Report

Authors have performed properly the changes required in the manuscript.

Congratulations for the research.